# UK nationals who received their medical degrees abroad: selection into, and subsequent performance in postgraduate training: a national data linkage study

Paul A Tiffin,[1] James Orr,[2] Lewis W Paton,[1] Daniel T Smith,[3] John J Norcini[4]

[1]Health Sciences, University of York, York, UK
[2]Emergency Department, Wythenshawe Hospital, Manchester University NHS Foundation Trust, Manchester, UK
[3]General Medical Council, London, UK
[4]Foundation for the Advancement of International Medical Education Research, Philadelphia, Pennsylvania, USA

**Correspondence to**
Dr Paul A Tiffin;
paul.tiffin@york.ac.uk

## ABSTRACT

**Objectives** To compare the likelihood of success at selection into specialty training for doctors who were UK nationals but obtained their primary medical qualification (PMQ) from outside the UK ('UK overseas graduates') with other graduate groups based on their nationality and where they gained their PMQ. We also compared subsequent educational performance during postgraduate training between the graduate groups.

**Design** Observational study linking UK medical specialty recruitment data with postgraduate educational performance (Annual Review of Competence Progression (ARCP) ratings).

**Setting** Doctors recruited into national programmes of postgraduate specialist training in the UK from 2012 to 2016.

**Participants** 34 755 UK-based trainee doctors recruited into national specialty training programmes with at least one subsequent ARCP outcome reported during the study period, including 1108 UK overseas graduates.

**Main outcome measures** Odds of being deemed appointable at specialty selection and subsequent odds of obtaining a less versus more satisfactory category of ARCP outcome.

**Results** UK overseas graduates were more likely to be deemed appointable compared with non-EU medical graduates who were not UK citizens (OR 1.29, 95% CI 1.16 to 1.42), although less so than UK (OR 0.25, 95% CI 0.23 to 0.27) or European graduates (OR 0.66, 95% CI 0.58 to 0.75). However, UK overseas graduates were subsequently more likely to receive a less satisfactory outcome at ARCP than other graduate groups. Adjusting for age, sex, experience and the economic disparity between country of nationality and place of qualification reduced intergroup differences.

**Conclusions** The failure of recruitment patterns to mirror the ARCP data raises issues regarding consistency in selection and the deaneries' subsequent annual reviews. Excessive weight is possibly given to interview performance at specialty recruitment. Regulators and selectors should continue to develop robust processes for selection and assessment of doctors in training. Further support could be considered for UK overseas graduates returning to practice in the UK.

### Strengths and limitations of this study

► The quantity, representativeness and completeness of the data available, consisting of 34 755 UK-based trainee doctors.
► The observational nature of the study meant we could not control for the effects of unmeasured variables not captured in the dataset.
► The use of Annual Review of Competence Progression (ARCP) as an outcome allowed comparisons to be made both across and within specialties.
► Some restriction of range may be present, since ARCP outcomes were only observed for those who entered specialty training.

## INTRODUCTION

The medical workforce is globalised, with international movement of doctors.[1] The UK is one of the largest net importers of doctors with around 33% of doctors registered with the medical regulator, the General Medical Council (GMC) having graduated from outside of the UK.[2] The situation is different for doctors in training, where in 2015, 85% have a UK Primary Medical Qualification.[3] Some specialties in the UK are particularly dependent on doctors who obtained their primary medical qualification abroad in order to fill training posts. Such specialties include general practice and the psychiatric specialties.[4] The reliance on overseas doctors in the UK is likely to continue. Indeed, the recent pledge by the Health Secretary for England, Jeremy Hunt, to provide 1500 extra medical school places per year, starting in 2018 will not provide additional applicants for basic specialty ('ST1 level') training programmes until 2025. This is based on a 5-year undergraduate medical degree and two subsequent postgraduate years of 'foundation' clinical training.[5] The effect of 'Brexit' (the

UK leaving the EU) is also likely to have an impact on the number of non-UK European doctors working in the National Health Service (NHS).

Previous research in the UK regarding international medical graduates (IMGs) has focused on identifying differences in performance on postgraduate exams and Annual Review of Competence Progression (ARCP) when compared with doctors with UK primary medical qualifications.[6–8] Such studies have identified a number of demographic and educational factors associated with later postgraduate academic performance. For example, IMGs who obtained higher scores on the Professional and Linguistic Assessments Board (PLAB) exams and the English fluency test (International English Language Testing System), used to obtain registration with the GMC, have educational outcomes closer to UK medical graduates (UKGs).[7 9 10] Differential attainment has also been observed in postgraduate medical examinations in North America.[11] There are a variety of views about the underlying reasons for these differences.[12]

Approximately 3%–4% of all UK doctors in training (see 'Results' section) are UK citizens who obtained their primary medical qualification outside of the UK—referred to here as 'UK overseas graduates (UK OGs)'. This includes those who obtained their medical degrees from both countries within and without the European Economic Area (EEA). At present, virtually nothing is known about this group of doctors. However, a previous study reported some interesting differences in performance on the PLAB exams between UK OGs and non-UK citizens who qualified outside of the EU (referred to here as 'IMGs'). Compared with the IMGs, the UK OGs were observed, on average, to have more attempts and lower scores on part 1 of the PLAB exam (the written component). In addition, they had reduced performance on the knowledge-based component of the Membership of the Royal College of General Practitioners exam (MRCGP) relative to the IMGs. Interestingly, no significant difference in scores on the clinical component of the MRCGP was observed between the two groups.[7] A North-American study reported that US citizens with non-US primary medical qualifications perform less well in the US Medical Licensing Examination and are less likely to be board-certified specialists compared with other groups of doctors.[13] Of more concern was the observation that the patients of doctors who are American citizens with non-US primary medical qualifications had poorer clinical outcomes than those treated both by non-US international and US medical graduates.[14] Thus, it is possible that differential educational attainment may translate, in some cases, to poorer clinical care.

For a doctor to practise in the UK, they must fulfil the requirements of the 1983 Medical Act.[15] For IMGs, this often involves evidencing their clinical competence by passing both parts of the PLAB test, although other routes to registration are available, especially for more experienced practitioners. The first part of the PLAB evaluates medical knowledge using multiple choice questions. Part 2 of the PLAB is an evaluation of practical clinical skills using a series of objective structured clinical examination stations. To be eligible to sit the PLAB test, doctors must have an acceptable medical degree.[15] Until 2014, for those from countries outside of the EEA, where English was not an official language, evidence of English proficiency needed to be provided. Since 2014, this applies to all countries, including EEA countries too, that are not on the GMC's 'first and native' English language list.[16] However, UK citizens who obtained their primary medical qualification outside of the EEA would generally have to pass the PLAB test in order to demonstrate clinical competency prior to obtaining a licence to practice. Paradoxically, this is not the case with citizens of other EEA countries who qualified from a non-European institution, as long as they have practised medicine within the EEA for at least 3 years. In this latter case, an exemption from sitting the PLAB test may be granted to the doctor via their 'enforceable community right', which is conferred via their European Union (EU) citizenship. Nevertheless, those seeking to register must provide robust evidence of their competence, and as no entitlement to registration exists under this route, failure to do so will result in refusal. This situation may change following the UK's exit from the EU.[17]

Obtaining a licence to practise does not guarantee employment. In particular, doctors registering to practise via the PLAB route will often be seeking to obtain a place on a specialty training programme, which is the most usual pathway to both general practice and senior hospital medical positions. In the UK, these are usually divided into core training and higher specialist training. It is usual for doctors to complete their core specialist training before applying for a higher specialist training post, although some training programmes, such as general practice, paediatrics and obstetrics and gynaecology are 'run through' and do not have such a break. The time spent at each stage will vary depending on the specialism. Postgraduate examinations, linked to the relevant Royal College, must be passed at varying stages in order to progress and eventually obtain a certificate of completion of training (CCT). This CCT permits a doctor to be placed on the GMC Specialist or GP Register. In the UK, recruitment into specialty training programmes is now largely organised around a nationalised system, although some local 'standalone' posts may still exist, particularly if they are short-term posts intended to cover unplanned vacancies (eg, 'Locum Appointed for Training' (LAT) posts'). Applications to specialist training programmes use the online Oriel system. Online application forms and evidence of qualifications are submitted and selectors then generate short-listing scores based on the job criteria. Applicants to some core-training posts, and for general practice, may also have to complete additional tests such as those found in the Multi-Specialty Recruitment Assessment (MSRA).[18] Applicants successful at any initial stages will be invited to attend further face-to-face assessments and interviews at national selection centres.[19] It is currently unknown,

other than for GP selection,[20] whether the probability of a doctor's success at specialty recruitment predicts, as ideally it should, subsequent educational performance in postgraduate medical training.

Previous research has identified a number of demographic factors that are associated with performance in postgraduate medical training.[8 21 22] From a patient perspective, the causes of any disparities in performance between medical graduate groups are not likely to be as important as the very fact they exist. However, attempting to control for the influence of such variables in studies of inter-group differences may be useful in clarifying the underlying relationships with the outcomes of interest. For instance, gender has been associated with performance in postgraduate medical examinations.[21] Increasing age tends to be associated inversely with performance both in postgraduate education[6 22] and practice.[23 24] In contrast, clinical experience tends to improve performance in both.[6 25] However, there is something of an interaction between age and experience in that increasing age tends to offset the benefits of experience when determining clinical outcomes.[23 26]

Qualitative research findings have suggested that linguistic and cultural factors may, at least partly, mediate these differential attainment rates between home and overseas medical graduates.[27 28] In UK citizens who graduate from abroad, these language and cultural factors may be assumed to be less prominent than in non-UK OGs. However, preparedness for practice has been highlighted as an issue even with UKGs.[29] It may be that those who experience their undergraduate training in another country may be less well prepared to work in the UK Health Services.[30] This may be reflected in educational performance. In particular, it is possible that those training in a very different socioeconomic context may be the most disadvantaged in this respect. Specifically, the nature of clinical practice may be shaped by the healthcare resources in a country.

The reasons for UK citizens applying to study medicine outside their home country are likely to be varied but at present unknown. However, in the USA roughly half of all Americans who study medicine outside of the States previously or concurrently applied to US medical schools.[31] Similarly, a major motivation for UK citizens applying abroad may be that they consider themselves unlikely to obtain a place to study medicine at a UK university. It is possible that family links, tuition costs and other sociocultural influences may encourage study outside of the UK.

Data are routinely collated by the GMC on demographics and educational performance of doctors registered to practise in the UK. Thus, the aims of our study were:

▶ To compare the likelihood of being deemed appointable to a national medical postgraduate training programme for UK citizens who obtain their medical degrees overseas, compared with other graduate groups.

▶ To evaluate whether subsequent differential attainment in postgraduate training was observed for such UK citizens who graduated abroad, compared with other groups of medical graduates.

▶ To compare any patterns observed, as above, in order to assess the effectiveness of selection into postgraduate training, in relation to a doctor's nationality and place of qualification.

Thus, our findings would have implications for both international medical regulators and employers.

## METHODS

Our aim was to compare the chances of success at specialty training selection between UK OGs and the other graduate groups, both before and after controlling for the effects of potential confounding factors, such as age, sex, and duration of UK-based experience. We then compared the subsequent ARCP outcomes across groups. Thus, we evaluated the extent to which the probability of success, in relation to other medical graduate groups, at selection was mirrored by subsequent ratings of performance in postgraduate training. The ARCP process involves a regular review of the progress of a UK doctor in training by an educational panel. This panel considers the evidence presented in the doctor's portfolio, which includes anonymised cases, reflections and feedback from a supervisor, colleagues and workplace-based assessments. It does not usually involve a face-to-face meeting unless issues arise that require clarification or a less than satisfactory outcome is likely to result.[32]

### Data sources and preparation

Data on the outcomes of recruitment to specialty training for the UK between 2012 and 2016 were obtained via an extract from the Oriel database[33] supplied to the GMC. The recruitment process was concerned with appointing doctors to training programmes at core-training (CT) and up to and including specialty training (ST) level 4 (ie, to the earlier years of a training programme, the length of which is determined by the specialty). The flow of data through the study is depicted in figure 1. Oriel recruitment data were potentially available for 52 894 doctors, of whom 34 755 were linked to the subsequent ARCP dataset (see below). The two potential outcome variables available from the Oriel recruitment database were '*deemed appointable*' and '*post offered*'. In order to reduce the impact of local competition on the results, the '*deemed appointable*' variable was used as the outcome measure for the recruitment data modelling. Data were also available on interview performance and shortlisting ratings, which were standardised as z-scores (mean 0, SD of 1) within each year and specialty. Note that in GP recruitment an applicant can receive an offer without interview if they score above a certain threshold on the MSRA.[18] In these cases, the interview score was treated as missing. In this sample, interview score was missing for 5198 applications for GP specialty training (17.63%).

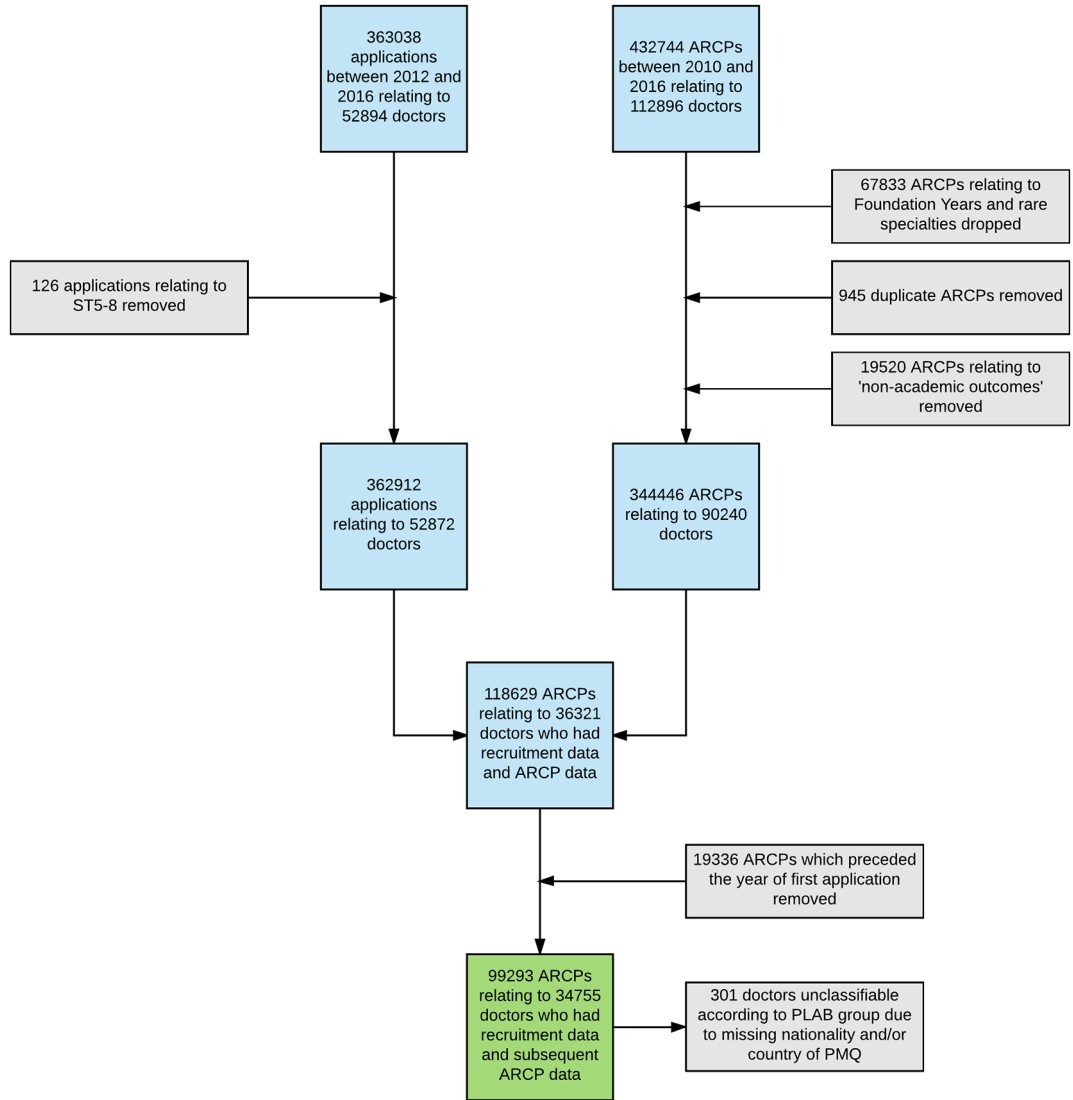

**Figure 1** Flow of data through the study. ARCP, Annual Review of Competence Progression; PLAB, Professional and Linguistic Assessments Board; PMQ, primary medical qualification.

Data relating to performance in training were potentially available for 90 240 doctors in specialist training with 344 446 competency-based ARCP outcomes recorded (see below and figure 1), who were in national postgraduate training schemes between August 2009 and August 2016. We also noted that there were 838 doctors (2.4%) who only had ARCP outcomes that were awarded in relation to short-term LAT or 'fixed-term specialty training appointment' posts. The data were analysed within a 'safe haven' environment.[34] The ARCP data are collected annually by the GMC and the collection notices are published on the GMC website.[35]

Record of in-training assessment scores were recoded to the equivalent ARCP outcome codes. Only ARCP 'competency-based' outcomes indicating training progress were included (eg, 'out of programme' experience was excluded). The remaining outcomes were then collapsed onto a 4-point ordinal scale:

► 1=*'satisfactory progression'/'training programme completed'* (ARCP outcomes '1' or '6', respectively);

► 2=*'additional evidence requested'* (ARCP outcome '5');

► 3=*'targeted training required (no extended time)'* (ARCP outcome '2');

► 4=*'extended training time required/left programme'*) (ARCP outcomes '3' or '4', respectively).

This was an approach previously shown to be valid.[6] The dataset also contained a variable, recorded by the deaneries, to indicate whether a specific ARCP outcome was associated with a failure to pass a postgraduate examination (ie, those required by the UK Royal Colleges as part of specialist training). When ARCP outcomes were treated as dichotomous, we classified any outcome other than a '1' ('satisfactory progress') or a '6' ('training programme completed') as 'less than satisfactory'. Note that this included outcome '5', which the ARCP 'Gold Guide' does not classify as 'unsatisfactory'.[32] Training deanery, specialty and medical training grade, all matched to individual ARCP records, were obtained from deaneries via the GMC.

**Table 1** Classification of medical graduate groups for purposes of study

| Group name | Region where PMQ obtained | Nationality |
|---|---|---|
| UK medical graduates | UK | All |
| EEA graduates | EEA | All—except for UK nationals |
| International medical graduates | Outside of EEA and UK | All—except for UK nationals |
| UK overseas graduates | Outside of UK (EEA or non-EEA country) | UK national |

EEA, European Economic Area; PMQ, primary medical qualification.

For the sample, the nationality and the name and country of the medical school where the primary medical qualification was obtained were derived from the GMC List of Registered Medical Practitioners. According to the GMC, dual nationality was recorded in 2115 (2.4%) of the sample. Where dual nationality occurred, only the first nationality provided by the GMC was used. The country of origin was deleted by the GMC only in four instances prior to release as a safeguard against identifying the individual doctors.

Graduate status was categorised predominantly, although not exclusively, according to GMC regulatory policy; that is, whether the doctor would have been expected to have passed the PLAB test in order to provide evidence of clinical competence and obtain a licence to practise. Thus, for analytic purposes the sample was grouped as follows:

1. *UKGs*—irrespective of country of nationality.
2. *Graduates who were nationals of countries from the EEA* (with the exception of the UK), irrespective of country of qualification.
3. *IMGs* who were non-EEA nationals, irrespective of place of graduation (except the UK).
4. *UK OGs*—UK nationals who graduated from an institution outside of the UK (either EEA or non-EEA).

The graduate group classification is further described in table 1.

Note that in the above classification that within the group of UK OGs there were those that would have been likely to sit the PLAB exams (ie, those who graduated outside of the EEA) and those that were not (ie, those graduating from universities within the EEA). For this reason, a subanalysis of this group was conducted (see 'Results' section). It should also be noted that group 2 (EEA graduates) was defined by nationality rather than place of qualification. This was because citizens of the EEA do not tend to obtain a licence via the PLAB route if they have worked clinically in a European country for 3 years or more. In our sample, 155 out of 1225 EEA nationals (12.65%) had obtained their primary medical qualification from outwith the EEA.

Dichotomised ethnicity data (white/non-white) were available from the GMC annual national training survey.[36] However, ethnicity was used for descriptive purposes only and not used in the modelling. This was because ethnic status served as a proxy for graduate group membership (eg, only 15% of the UK OGs reported themselves as of white ethnicity). Thus, ethnicity was, in effect, confounded by graduate group allocation. Sex, year of birth and date of first UK medical registration was also obtained from the List of Registered Medical Practitioners. For the recruitment data analyses, the duration of experience in UK medical practice was calculated from the years of birth and application. For analyses relating to ARCP, the duration of experience in UK medical practice was calculated from the year of birth and date of ARCP.

Specialties were classified predominantly according to the Royal College they were affiliated to, as in our previous study.[6] We wished to understand whether training in a relatively less well-resourced undergraduate medical environment mattered or whether the degree of dissonance between the country of nationality and country of qualification was relevant. Consequently, we derived a metric of the economic status of the country of nationality and qualification, and the difference between these two for each doctor. This was done by linking the gross domestic product (GDP) per capita in US dollars, according to the 2008 World Bank data[37] to the name of the associated countries in the sample. The discrepancy between these values was also calculated as both a relative and absolute difference.

### Patient and public involvement

Patients were not involved in this study.

### Analyses

For the recruitment outcomes, multilevel logistic regressions were used to estimate predictive models for the binary outcomes. Application events were treated as repeat measurements nested within doctors, with the intercept of the model allowed to vary randomly across each applicant.

Likewise, ARCPs were treated as repeat measurements nested within doctors. Thus, multilevel ordinal logistic regression analyses were used to estimate the odds of obtaining a less versus a more satisfactory ARCP outcome. The intercept of the model was allowed to vary randomly across each doctor in training. No clustering effects (as indicated by the intraclass correlation) for deanery were observed and thus no control for this was required. For the prediction of subsequent training performance, analyses were conducted both with and without ARCP outcomes associated with postgraduate exam failure. This was in order to evaluate the impact of the examination performance on the ARCP outcomes in each graduate group and across the main medical specialties.

For both sets of analyses, the baseline category of graduate group was swapped to evaluate all combinations of comparison. As part of the modelling process, all

combinations of interactions between the predictor variables were evaluated. Only interaction terms that were statistically significant (at the p<0.05 level) and substantively meaningful were included in the final models. Multivariable model building proceeded in a forward stepwise manner with a p value of <0.05 from univariable analysis being the criterion for entry. The predictor variables used in the multivariable model building, including the available potentially confounding variables, were: age, UK experience, ethnicity, sex, selection standardised shortlisting scores and standardised interview scores.

Analyses by specialty group were conducted and the results from three of these reported as exemplars: *general practice*, *psychiatry* and *surgery*. General practice and psychiatry were selected as they had relatively high proportions of UK OGs working in them. Moreover, relatively high differential performance at both ARCP and postgraduate membership exams between UK and international graduates have been reported in these latter two specialties.[6–8] The results for the surgical specialties are also presented as, traditionally, entry to the training schemes are more competitive than most other medical fields.[38]

Missing data were relatively uncommon (figure 1), other than for shortlisting score and interview score. As such, we repeated the above analyses using multiply imputed data using chained equations, creating 20 imputed datasets, as implemented in STATA 14. This portion of the analysis can be thought of as sensitivity analyses for these two selection variables. Specifically, if the results between the imputed and non-imputed datasets vary then this would be evidence that the absent values are 'missing not at random' (ie, the missing values are neither associated with the observed data nor due to chance). Thus, the results in relation to any affected variables must be interpreted more cautiously.

## RESULTS
### Descriptive statistics
As can be seen from figure 1, there were relatively few missing data in the final sample of doctors. For example, country of nationality and/or qualification was unavailable in only 301 (0.8%) of the doctors in this final sample. The exceptions to this are the shortlisting score, which was missing in 45.9% of cases, and interview score, unavailable for 30.2% of the final sample.

Table 2 shows the demographic characteristics of the doctors in the final sample (where both recruitment and ARCP outcomes were available) in relation to the recruitment outcomes. As can be seen from table 2, UK OGs were more likely than UK graduates to be male and report non-white ethnicity. It can also be seen that, on average, UK OGs had slightly lower standardised shortlisting scores but somewhat higher mean interview scores than non-UK IMGs. UK OGs were, on average, approximately 5 years older than UK graduates at specialty application with around a year of extra UK clinical experience at the time of the first recorded ARCP. It can also be seen from table 2 that, on average, IMGs applied for more specialty posts than UK OGs during the study period, although were less often deemed appointable by the selection panel. The background characteristics and overall ARCP outcomes for the four groups of medical graduates are shown in table 3. As can be seen, compared with other graduate groups, UK OGs were more likely to receive a 'less than satisfactory' outcome at ARCP, which was more likely to be associated with a failure to pass a postgraduate exam.

Table 4 depicts a breakdown of the composition of doctors in training in each specialty according to graduate group. Overall 1108 (3.19%) of the doctors in our sample were UK OGs. As can be seen, those disciplines where competition for training places is less competitive[38] tend to have the highest proportion of UK OGs, such as psychiatry (145/2183, 6.6%).

### Modelling outcomes from specialty selection
The results from the univariable analyses are depicted in table 5 and figure 2. In order to further reduce the impact of competition effects (ie, less highly achieving candidates applying for the least competitive specialties), we also repeated the specialty-based analyses with the

**Table 2** Background and specialty recruitment characteristics of the doctors in the sample by graduate group

| Group name | Male (%) | Non-white ethnicity (%) | Mean shortlisting z-score (SD) | Mean interview z-score (SD) | Mean no. of jobs applied to during the study period (SD) | Mean no. deemed appointable (SD) | Mean no. offered (SD) |
|---|---|---|---|---|---|---|---|
| UK medical graduates | 12 008/28 293 (42%) | 9235/28 249 (33%) | 0.31 (0.84) | 0.27 (0.91) | 1.97 (1.41) | 1.36 (0.87) | 1.25 (0.74) |
| EEA graduates | 535/1225 (44%) | 191/1209 (16%) | −0.17 (1.06) | −0.24 (0.93) | 2.49 (2.13) | 1.17 (1.00) | 0.96 (0.80) |
| International medical graduates | 2120/3828 (55%) | 3597/3766 (96%) | −0.46 (0.91) | −0.45 (0.90) | 3.08 (2.66) | 1.01 (0.98) | 0.80 (0.73) |
| UK overseas graduates | 652/1108 (59%) | 925/1093 (85%) | −0.45 (0.90) | −0.42 (0.92) | 2.94 (2.49) | 1.11 (0.91) | 0.92 (0.71) |

Recruitment data values for the study period August 2009–2016. EEA, European Economic Area.

**Table 3**  Summary of the background and Annual Review of Competence Progression (ARCP) descriptive statistics for the doctors in the study sample

| Group name | Mean age at first ARCP (SD) | Mean UK experience at first ARCP (SD) | Mean number of ARCPs (SD) | Proportion 'unsatisfactory' (ie, not outcome 1 or 6) | Proportion of ARCPs associated with postgraduate exam failure (%) |
|---|---|---|---|---|---|
| UK medical graduates | 29.00 (3.31) | 2.62 (1.30) | 3.29 (2.04) | 19 511/80 361 (24.28%) | 3213/80 361 (4.00%) |
| EEA graduates | 31.85 (4.55) | 3.54 (2.06) | 3.22 (2.04) | 1093/3341 (32.71%) | 347/3341 (10.39%) |
| International medical graduates | 34.68 (4.66) | 4.14 (2.08) | 3.70 (2.38) | 3902/11 404 (34.22%) | 1086/11 404 (9.52%) |
| UK overseas graduates | 34.38 (5.60) | 3.82 (2.10) | 3.49 (2.15) | 1167/3174 (36.77%) | 348/3174 (10.96%) |

EEA, European Economic Area.

three exemplars (*psychiatry, surgery* and *general practice*). Results by specialty are shown in figure 3. As can be seen from the results in the left hand column of table 5 and also figure 2, UK OGs were less likely than UK graduates or EEA nationals to be deemed appointable at specialty selection. However, they were more likely than IMGs who were not UK nationals to be deemed appointable (OR 1.29, 95% CI 1.16 to 1.42). Also apparent in table 5 is that females, younger and more experienced doctors were more likely to be successful at recruitment. Those who were nationals or qualified from wealthier countries were also more likely to be deemed appointable. Disparities between a country of qualification and nationality's income, in either direction, were associated with a reduced odds of being deemed appointable, with the

odds being roughly reduced by 25% for every difference of US$10 000 per capita.

The results from the multivariable analyses, predicting outcome at specialty recruitment are also presented in table 5, in the right-hand column. Results adjusted for various predictor variables are also shown in figure 2. One interaction term that was statistically significant at the $p<0.05$ level and conceptually justifiable was included in the modelling. This was the term representing the interaction between age and experience. This term, when exponentiated as an OR, was <1. This highlights that the advantage of increasing experience at interview was offset by more advanced age. The pattern in the multivariable results observed was generally similar to the univariable results. As can also be seen from figure 2, as expected,

**Table 4**  Number of doctors in the sample (percentage of total) in specialty training by graduate group and specialty

| Specialty group | UK medical graduates (%) | EEA nationals (%) | IMGs (%) | UK overseas graduates (%) |
|---|---|---|---|---|
| Anaesthetics | 2918 (91.53) | 61 (1.91) | 149 (4.67) | 37 (1.16) |
| Medicine | 6329 (79.18) | 336 (4.20) | 1040 (13.01) | 233 (2.92) |
| Psychiatry | 1289 (61.82) | 143 (6.86) | 489 (23.45) | 136 (6.52) |
| Surgery | 3394 (82.82) | 173 (4.22) | 357 (8.71) | 131 (3.20) |
| Emergency medicine and acute care common stem | 1070 (80.21) | 47 (3.52) | 162 (12.14) | 46 (3.45) |
| General practice | 9353 (83.45) | 219 (1.95) | 1126 (10.05) | 409 (3.65) |
| Obstetrics and gynaecology | 822 (78.36) | 51 (4.86) | 132 (12.58) | 36 (3.43) |
| Occupational medicine | 6 (50.00) | 1 (8.33) | 3 (25.00) | 0 (0.00) |
| Ophthalmology | 341 (82.37) | 26 (6.28) | 29 (7.00) | 10 (2.42) |
| Paediatrics | 1557 (81.82) | 105 (5.52) | 192 (10.09) | 37 (1.94) |
| Lab based | 309 (75.55) | 23 (5.62) | 64 (15.65) | 11 (2.69) |
| Public health | 123 (91.79) | 3 (2.24) | 1 (0.75) | 1 (0.75) |
| Radiology | 688 (83.29) | 36 (4.36) | 78 (9.44) | 20 (2.42) |
| All specialties | 28 199 (81.41) | 1224 (3.52) | 3822 (11.01) | 1108 (3.19) |

EEA, European Economic Area; IMG, international medical graduates.

**Table 5** Results from univariable and multivariable multilevel logistic regressions predicting the odds of being deemed appointable at specialty training recruitment

| Predictor | Univariable models<br>OR (95% CI) | Multivariable model<br>OR (95% CI) |
|---|---|---|
| UK OGs vs UKGs | 0.25 (0.23 to 0.27) | 0.65 (0.56 to 0.77) |
| UK OGs vs EEAG | 0.66 (0.58 to 0.75) | 0.65 (0.53 to 0.81) |
| UK OGs vs IMG | 1.29 (1.16 to 1.42) | 1.05 (0.89 to 1.23)* |
| Male sex | 0.70 (0.68 to 0.73) | – |
| Shortlisting score (z-score) | 1.74 (1.69 to 1.80) | – |
| Interview score (z-score) | 8.41 (8.10 to 8.73) | 6.78 (6.47 to 7.10) |
| Age at selection | 0.93 (0.93 to 0.94) | – |
| UK experience (years) at selection | 1.50 (1.47 to 1.52) | 1.17 (1.08 to 1.27) |
| Experience/age interaction | – | 0.99 (0.99 to <1.00)† |
| GDP of country of nationality (US$10 000 per person) | 1.52 (1.50 to 1.55) | – |
| GDP of country of qualification (US$10 000 per person) | 1.63 (1.60 to 1.65) | – |
| Difference in GDP countries (US$10 000 per person) | 0.87 (0.85 to 0.89) | – |
| Absolute difference in GDP (US$10 000 per person) | 0.74 (0.73 to 0.76) | – |

EEAG, European Economic Area graduate; GDP, gross domestic product; IMG, international medical graduate; UK OG, UK overseas graduate; UKG, UK medical graduate.
*p>0.05
†p=0.01

progressively controlling for background variables, shortlisting scores and interview performance reduces the disparities in odds of being deemed appointable between the UKGs and non-UK graduate groups. Likewise, it can be seen from the results shown in table 5 that the difference in the odds between a UK overseas graduate and

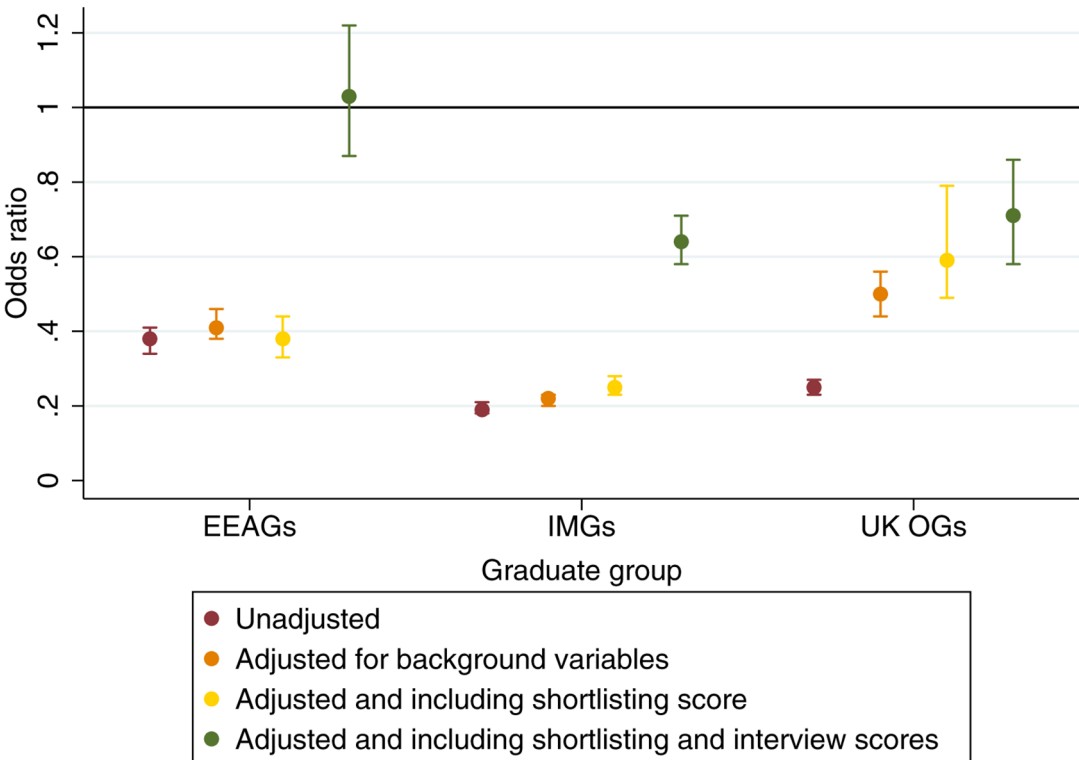

**Figure 2** Results from univariable and multivariable analyses for an individual being deemed 'appointable' for each graduate group in comparison to UK graduates. EEAG, European Economic Area graduate; IMG, international medical graduate.

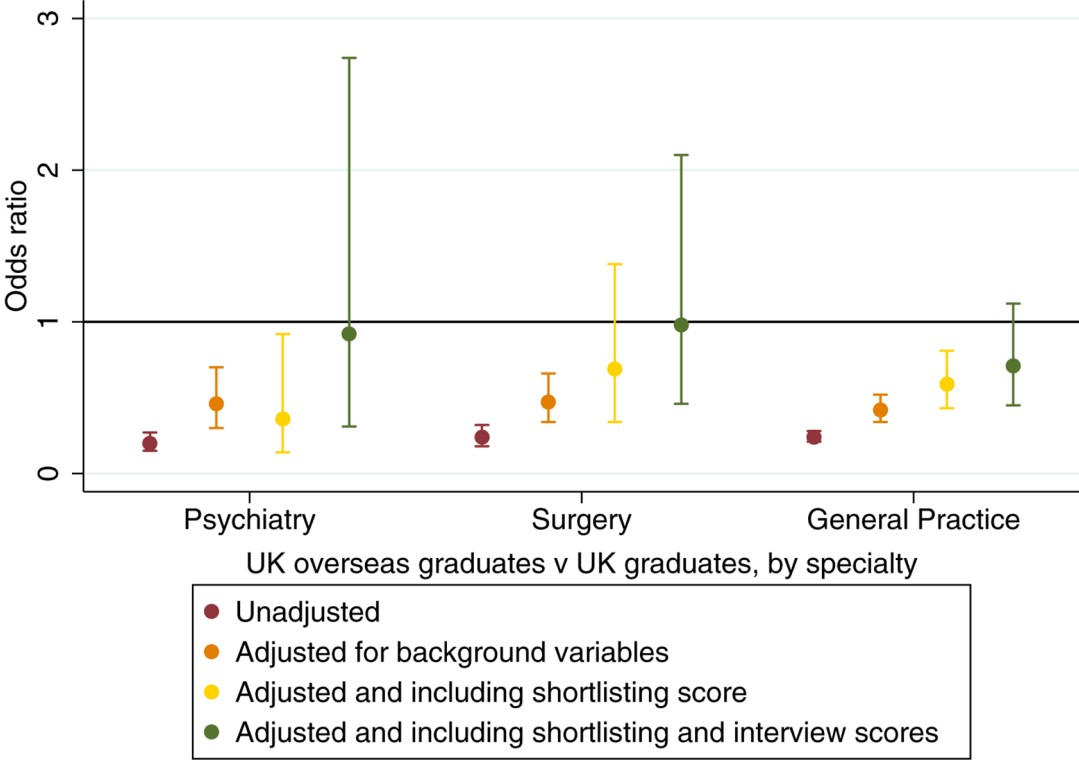

**Figure 3** Results by specialty for UK overseas graduates being deemed 'appointable' vs UK graduates.

a non-UK IMG also decreased after controlling for the influence of other predictor variables. Another noteworthy observation is that once the odds are conditioned on the relative interview scores the shortlisting scores become non-significant predictors of 'appointability'.

### Modelling ARCP outcomes

In total, there were data relating to 99 293 ARCP outcomes relating to 34 755 doctors in specialist training in the final dataset. As can be seen from the flow of data depicted in figure 1, there were relatively few missing data. Statistical significance, in the present case, should be assumed to be at p<0.001 level, unless otherwise stated.

In terms of univariable analyses, graduate group was associated with an increased odds of receiving a less satisfactory ARCP outcome at review, with UK OGs showing the largest difference with UKGs as the comparison category (figure 4). This pattern was replicated across the three specialties selected as exemplars (figure 5). We also noted no significant difference in the odds of a less satisfactory ARCP outcome between UK OGs who are generally expected to sit the PLAB test (n=812) and those who had qualified from a medical school within the EEA (n=793, OR 1.16, 95% CI 0.91 to 1.47, p=0.24); the latter subgroup (n=294) being exempt from the test.

Both increasing age and UK experience at ARCP were associated with higher odds of a less satisfactory outcome, as was male sex. In the case of age and experience, the odds of a poorer versus better outcome increased by approximately 5% per year. Likewise, the odds of males

having a less, rather than more, satisfactory outcome, were about 43% higher than those for females doctors (table 6).

Both the GDP of the country of nationality and that of the place of qualification had roughly equal influences on ARCP outcomes: for every US$10 000 extra per capita the GDP of the country the odds of a less satisfactory (vs more satisfactory) outcome dropped by approximately 20%. However, the most potent predictor in this regard was the absolute difference in GDP between country of nationality and place of qualification. This indicated that the odds of a less satisfactory (vs more satisfactory) outcome increased by approximately 22% for every US$10 000 per capita difference between the two countries, regardless of the direction of the disparity.

Once ARCP outcomes associated with postgraduate exam failure were excluded from the analyses, the effect sizes of the predictors diminished to varying degrees, although remained statistically significant in all cases (table 6—right hand column and figure 4). This indicated that some of the association between ARCP outcomes and graduate status and the other predictors were mediated by differential Royal College exam pass rates.

The influence of background variables was controlled for in the multivariable analyses. We observed that the differences in ARCP outcomes between the UKGs and those who held non-UK primary medical qualifications diminished to some extent (figure 4). Moreover, when the impact of differential postgraduate exam pass rates

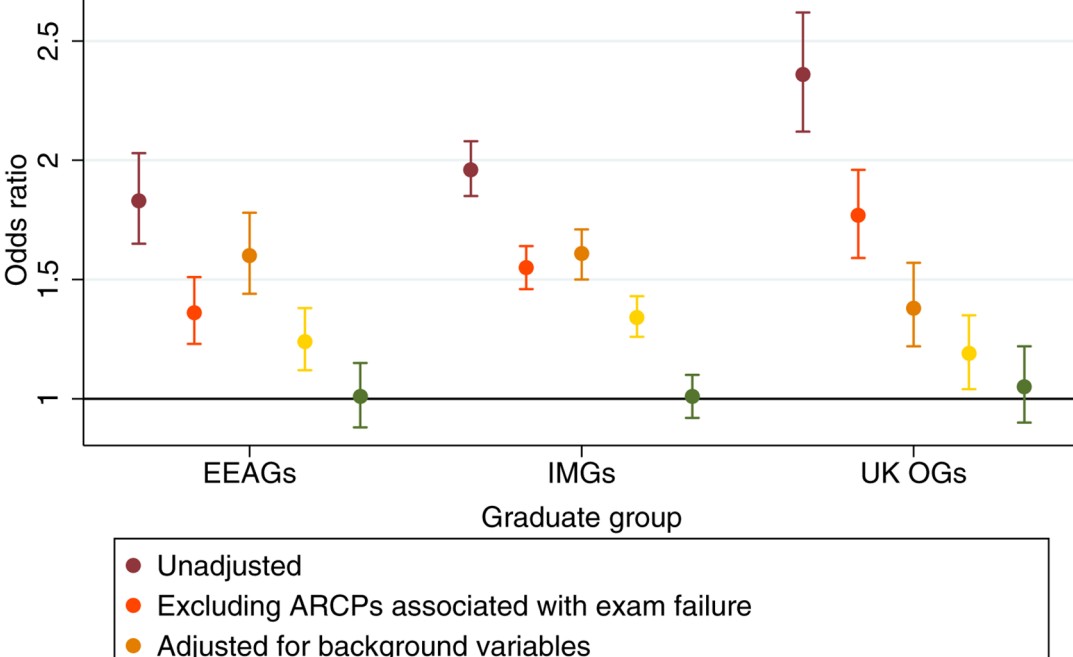

**Figure 4** Univariable and multivariable ARCP results for all specialties for each graduate group in comparison to UK graduates. ARCP, Annual Review of Competence Progression; EEAG, European Economic Area graduate; IMG, international medical graduate; UK OG, UK overseas graduate.

were also adjusted for (by excluding the relevant ARCP outcomes), the intergroup differences further reduced, disappearing entirely for UK OGs versus home medical graduates. We also noted that (in contrast to the univariable results) once the influence of age was controlled for, UK-based experience predicted the probability of more, rather than less, satisfactory ARCP outcomes. The results are also depicted in table 7.

The picture when analyses were conducted for each specialty group was similar (figure 5). As can be seen, even after adjusting only for the influence of background variables there were no differences remaining in the odds of a more satisfactory ARCP outcome between UK overseas and home graduates in the surgical trainees. Likewise, no statistically significant differences remained after excluding ARCPs associated with exam failure in psychiatry and general practice (figure 5). It is worth noting that all the other predictors in the model, including an interaction term for age and experience, remained statistically significant and independent predictors of ARCP outcome (table 7).

### Results from the imputed datasets

When re-running the multivariable analysis predicting 'appointability' at specialty recruitment on the imputed data, unlike the analysis on the non-imputed dataset (table 5), both shortlisting score (OR 1.05, 95% CI 1.03 to

1.08) and interview score (OR 3.13, 95% CI 3.03 to 3.23) are significant independent predictors.

In contrast, the analyses for ARCP outcome show negligible difference whether being performed on the non-imputed data or the imputed data.

### DISCUSSION

In this study, the first to focus on UK OGs, we observed that this group of doctors were more likely to be deemed appointable at specialty training recruitment than non-UK IMGs, although less so than other graduate groups. Marked disparities in ARCP outcomes between this group of doctors and other graduate types were also noted. The patterns observed in the selection data were not precisely replicated, with a 're-ordering' of UK and non-UK overseas medical graduates. In the present case, the dissonance between the selection and ARCP results could be largely, if not wholly, explained by the differential interview performances between UK and non-UK IMGs. This finding is consistent with a previous report into selection and subsequent educational achievement in those recruited to UK general practice training— performance on selection measures were noted to be less strongly predictive of subsequent scores at the MRCGP exam in international, compared with home graduates.[20]

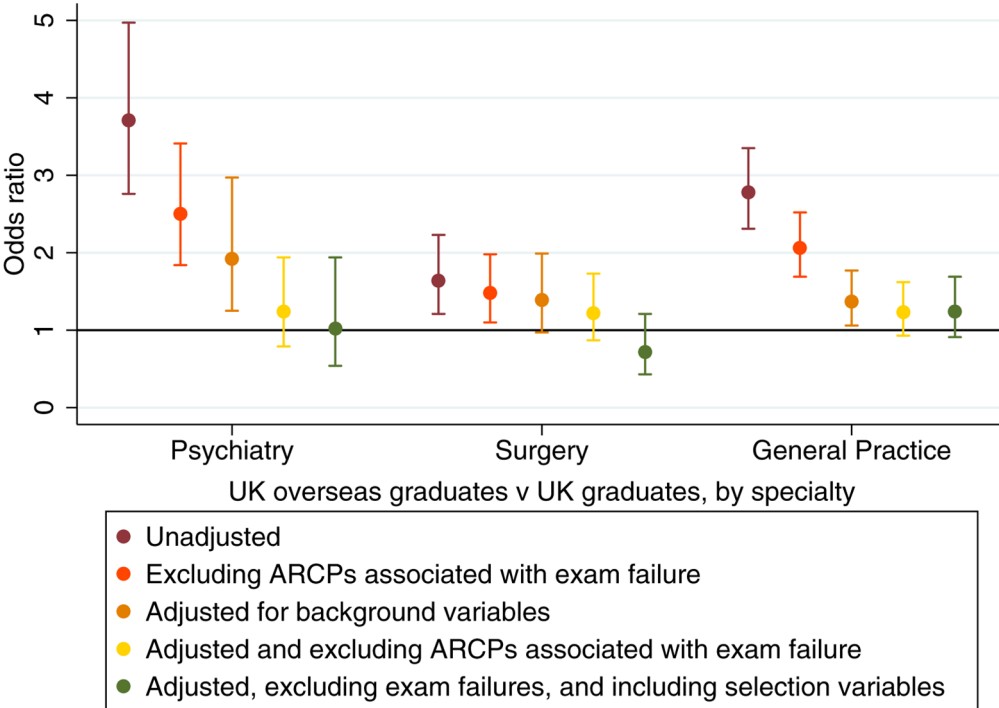

**Figure 5** Univariable and multivariable odds ratios for receiving a less satisfactory vs a more satisfactory ARCP outcome. The results are shown only for UK overseas graduates vs UK graduates, by specialty. ARCP, Annual Review of Competence Progression.

The magnitude of intergroup differences in ARCP outcomes reduced after controlling for the influence of age, UK experience and absolute economic differences between the country of nationality and qualification. We also observed a significant interaction between age and (UK-based) clinical experience, in line with previous

**Table 6** Results from a series of univariable multilevel ordinal logistic regression analyses predicting the odds of 'less' vs 'more' satisfactory ARCP outcomes for the sample of doctors (n=34 755)

| Predictor | Including exam failures OR (95% CI) | Excluding exam failures OR (95% CI) |
|---|---|---|
| UK OGs vs UKG | 2.36 (2.12 to 2.62) | 1.77 (1.59 to 1.96) |
| UK OGs vs EEAG | 1.29 (1.12 to 1.48) | 1.30 (1.13 to 1.50) |
| UK OGs vs IMG | 1.20 (1.07 to 1.35) | 1.14 (1.02 to 1.28)* |
| Male sex | 1.43 (1.38 to 1.49) | 1.37 (1.32 to 1.43) |
| Age at ARCP | 1.05 (1.05 to 1.05) | 1.03 (1.03 to 1.04) |
| UK experience (years) at ARCP | 1.04 (1.03 to 1.05) | 1.02 (1.02 to 1.03) |
| GDP of country of nationality (US$10 000 per person) | 0.83 (0.82 to 0.84) | 0.88 (0.87 to 0.90) |
| GDP of country of qualification (US$10 000 per person) | 0.80 (0.79 to 0.82) | 0.87 (0.85 to 0.88) |
| Difference in GDP countries (US$10 000 per person) | 1.06 (1.03 to 1.09) | 1.03 (>1.00 to 1.05)† |
| Absolute difference in GDP (US$10 000 per person) | 1.22 (1.19 to 1.25) | 1.15 (1.12 to 1.18) |
| Mean shortlisting score (z-score) | 0.63 (0.61 to 0.64) | 0.76 (0.74 to 0.78) |
| Mean interview score (z-score) | 0.61 (0.59 to 0.62) | 0.71 (0.69 to 0.72) |

In the right hand column, the results from analyses where ARCP outcomes associated with postgraduate exam failure were excluded are shown.
*p=0.02.
†p=0.04.
ARCP, Annual Review of Competence Progression; EEAG, European Economic Area graduate; GDP, gross domestic product; IMG, international medical graduate; UK OG, UK overseas graduate.

**Table 7** Results from two multivariable multilevel ordinal logistic regression analyses predicting the odds of 'less' vs 'more' satisfactory ARCP outcomes for the sample of doctors (n=34 755)

| Predictor | Including exam failures OR (95% CI) | Excluding exam failures OR (95% CI) |
|---|---|---|
| UK OGs vs UKG | 1.08 (0.92 to 1.26)* | 1.05 (0.90 to 1.22)† |
| UK OGs vs EEAG | 0.96 (0.79 to 1.16)† | 1.04 (0.86 to 1.26)† |
| UK OGs vs IMG | 1.03 (0.88 to 1.21)† | 1.03 (0.88 to 1.21)† |
| Male sex | 1.28 (1.22 to 1.34) | 1.29 (1.23 to 1.35) |
| Age at ARCP | 1.02 (1.01 to 1.03)‡ | >1.00 (0.99 to 1.01)† |
| UK experience (years) at ARCP | 0.84 (0.79 to 0.90) | 0.83 (0.78 to 0.88) |
| Age/experience interaction | >1.00 (>1.00 to 1.01) | >1.00 (>1.00 to >1.00) |
| Absolute difference in GDP (US$10 000 per person) | 1.06 (1.02 to 1.10)§ | 1.05 (1.01 to 1.09) |
| Mean shortlisting score (z-score) | 0.72 (0.69 to 0.74) | 0.84 (0.81 to 0.86) |
| Mean interview score (z-score) | 0.67 (0.65 to 0.69) | 0.74 (0.72 to 0.77) |

In the right hand column, the results from analyses where ARCP outcomes associated with postgraduate exam failure were excluded are shown.
*p=0.3.
†p>0.5.
‡p=0.002.
§p=0.001.
ARCP, Annual Review of Competence Progression; EEAG, European Economic Area graduate; GDP, gross domestic product; IMG, international medical graduate; UK OG, UK overseas graduate.

findings.[23] In addition, the intergroup differences further diminished after excluding ARCP outcomes associated with exam failure, and indeed vanished for UK OGs versus home graduates following these adjustments. This pattern was generally seen across the medical specialties.

**Comparison with other studies and possible interpretations**
Our observation that the disparities in the odds of entering specialist training in the UK recruitment data were not precisely mirrored by ARCP outcomes was consistent with data from North America on selection to medical specialties. The US IMGs tend to have higher odds of being placed in a residency programme than their non-US counterpart,[39] despite subsequent reduced specialty certification rates for North American citizens who trained outside of the country.[40]

Likewise, our findings in relation to postgraduate educational performance were in keeping with those from previous studies of postgraduate education performance in IMGs training in the UK.[10] Specifically, our observations concurred with those reported by McManus and Wakeford who reported lower scores on the PLAB part 1 (written component) and Applied Knowledge Test of the MRCGP exam.[7] However, in the latter study only a subset of UK IMGs (ie, those who registered via the PLAB system) were included. We noted that the magnitude of intergroup differences was less marked in the more competitive disciplines. Over time, increasing competition ratios may drive up educational performance and so reduce any disparities between medical graduate groups.

By comparing the raw and adjusted ORs we obtain some indications of factors that may underlie the observed differences in ARCP outcomes. Certainly, age, UK experience

and the interaction between these two variables play a role. Moreover, it appeared that it was the absolute, rather than relative, difference between the economic status of country of nationality and qualification which had the larger influence on ARCP outcomes. This observation leads to the inference that there was something different about those individuals who studied medicine in a setting economically, and probably culturally different to their home country. It is also interesting to observe that controlling for the influence of the background predictors and postgraduate exam pass rates reduced the difference in ARCP outcomes between UK overseas and home graduates to a somewhat greater degree than those between the latter group and non-UK IMGs (figure 4). It is thus likely that some of the remaining, unexplained gap in ARCP performance between these latter two groups is accounted for by, perhaps subtle, linguistic and sociocultural factors, previously referred to as the 'dark variance' of differential attainment.[41]

It also seemed that many of the 'less than satisfactory' ARCP outcomes (as defined by the authors) were disproportionately associated with exam failure in UK IMGs. Thus, removing these reviews from the analyses diminished the observed intergroup differences. This raises questions about both the reliability of each assessment, as well as the constructs that they purport to measure. At present, the reliability of the ARCP is unknown, although the process has recently been subject to a qualitative review.[42] In contrast, there is some existing evidence of acceptable reliability for the Royal College postgraduate examinations.[43 44] Moreover, where psychometrically investigated, no evidence of racial bias was detected[45]

though, at least for the MRCGP CSA, more subtle socio-cultural forms of bias cannot be ruled out.[8 27] It could also be assumed that ARCP panels consider a range of factors in addition to clinical knowledge and skills, although do not usually involve a face-to-face interview.[32] These attributes may include perceived professionalism, ability to team work and administrative efficiency. Regarding the contrast between ARCP and the recruitment into specialty training results, it could be argued that the latter process gives some additional scope for bias (both conscious and unconscious). For example, there is evidence that face-to-face interviewers sometimes base decisions on misleading cues.[46] Although there are structured elements to the selection process, it can be assumed that 'softer' abilities, such as presentational skills will partly determine the outcome.[47] Thus, non-UK candidates, and especially those for whom English is not their first language, may be disproportionately disadvantaged, compared with the ARCP process. This possibility is supported by our observation that UK OGs received, on average, higher interview scores at selection than non-UK IMGs. It was noteworthy that once performance at interview was controlled for in the modelling, the graduate groups restacked into an order that was more consistent with that observed for ARCP performance. This raises the issue about whether excessive weight is given to interview performance within the specialty selection process. Such 'overweighting' might lead to situations where a candidate destined for satisfactory performance in postgraduate training is passed over in favour of one who outperforms them at interview but is less likely to make satisfactory future progress. It should be recognised that both in the UK and elsewhere postgraduate medical selectors are working to increase the standardisation and structure of their processes.[48] In particular, the introduction of Situational Judgement Tests as a component of selection into UK general practice training may effectively evaluate some of the non-academic qualities of candidates.[49] Such assessments are usually used in early stage screening processes, which leaves scope for candidates to diverge in performance at later stages of selection. Thus, these observations raise the question of whether ARCP and selection processes should become more like the postgraduate exams or vice versa? Perhaps ideally postgraduate examinations should test a wider range of qualities important to real-world practice, including the ability to demonstrate culturally appropriate professionalism and teamworking. Likewise, the ARCP process has been recently qualitatively reviewed[41] and at the time of writing there are plans aimed at improving its reliability, acceptability and validity.[50]

## Strengths and potential limitations

The primary strength of this study is the quantity, representativeness and completeness of the data available. This leads to the power to detect intergroup differences, even in subgroup analyses. The use of ARCP as an outcome allowed comparisons to be made both across and within specialties. We were also able to adjust for the impact of differential pass rates at postgraduate exams. Unlike in some previous studies, country of both nationality and place of qualification were available, allowing a more granular analysis, including by GDP. However, it should be noted that we only had access to the nationality of the doctor at the point of registration with the GMC. Thus, we were unable to differentiate between doctors who were designated UK citizens at birth and those who obtained this status subsequently.

The major limitation in this case was the observational nature of the study. Thus, we could not control for the effects of unmeasured variables not captured in the dataset. Nevertheless, by controlling for the effects of the predictor variables we had access to, as well as by excluding ARCPs associated with postgraduate exam failure, we were able to obliterate the observed differences in overall performance between the UK overseas and home graduates. Naturally, these results do not give rise to causal explanations for the differences. However, they do guide the focus of further investigation into the factors underlying these disparities, for example, differential pass rates at the Royal College membership examinations. We also noted a small percentage (2.4%) of doctors who only had ARCP outcomes recorded in relation to short-term training posts (eg, LATs). The posts held by this small group of doctors may have not been typical of training posts in general. However, when we excluded these medical trainees from the analyses no meaningful impact on the results was noted. Moreover, in practice, such short-term training appointments are sometimes awarded to doctors who then subsequently obtain a place on a substantive training programme. Thus, it did not appear practicable to differentiate between such temporary posts and long-term training programmes in the analyses.

It should be noted that, in this study, ARCP outcome '5' ('further information required') was used as an intermediate outcome category when conducting our modelling. In practice, a request for further information may, occasionally, be due to the failure of a supervisor, or other third-party, to supply documentation and may not be due to the actions of the trainee themselves. However, in line with our previous findings and exploration of the use of ARCPs as an educational outcome, it was felt that use of the 'outcome 5' in this way was justified.[6]

When comparing the results from the analysis of ARCP outcomes with the recruitment processes, it must be borne in mind that some level of 'filtration' has already occurred by the time doctors enter postgraduate training. That is to say that the range of the data has been restricted in that ARCP outcomes were only observed for those doctors successfully entering specialty training. Thus, the degree of disparity between the different graduate groups may have been underestimated. In particular, those who obtained their primary medical qualification outside of the UK may be especially likely to be in non-training medical posts and not included in the present sample.

Moreover, at least for the more competitive specialties, in accordance with EU employment law, those who were not nationals from the EEA may not have been shortlisted if there were deemed sufficient numbers of applicants from Europe. In addition, we only had access to date of registration with the GMC and could not estimate years of practice outside of the UK. However, it may be that practice in a comparable healthcare setting may be more important in predicting educational and clinical performance than experience per se. A further limitation was that ARCPs were not directly linked to the programmes interviewed for. This was because the structure of the data were complicated and the doctors sometimes changed specialty, or were undergoing 'dual' training in more than one specialty.

Some caution must be exercised when interpreting the interview scores as predictors of recruitment outcomes. The decision to deem a candidate 'appointable' is almost wholly based, at that stage of selection, on the interview ratings, and thus there is a tautological element to this aspect of the analysis. Nevertheless, it was informative to compare the standardised interview scores between the graduate groups. This permitted us to identify the source of the advantage that the UK OGs had over the non-UK citizens who had graduated from outside of the EEA at selection.

Both shortlisting score and interview score displayed extensive missingness, and the analysis relating to selection into specialty training on the imputed dataset produced somewhat differing results than those for the non-imputed data. The reduction in OR observed for the interview score in the imputed data set is not an unexpected result—only an application which proceeded to interview will have an associated interview score. Thus, there is likely to be some 'restriction of range' present. The shortlisting score was missing in nearly half of all cases, and the change in significance for shortlisting score in the multivariable model suggest that these data were not missing at random. It is possible that the missing data were at least partly due to differences in deanery returning practices. As the imputed analysis displayed modestly different results, some caution must be exercised when interpreting the results specifically relating to appointability at specialty recruitment and the shortlisting scores.

These findings raise important questions that could be answered by both further quantitative and qualitative studies. If the data could be made available, further research could be conducted to understand the differences in Royal College examination performance across the four graduate groups that the ARCP results presented here suggest exist. The reasons why UK citizens study abroad may be varied; ideally one could identify whether any such individuals had applied to medical school in the UK unsuccessfully or attended a UK medical school but left at some point. For these cases, it may be possible to obtain data from their application to UK medical school in the UK, for example, aptitude test scores, as a measure of their educational performance prior to completing their degree.

## CONCLUSIONS

We observed a significant effect for NHS experience. This implies that in order to enhance the postgraduate educational performance of doctors who graduate from overseas, additional training opportunities could be effective. In particular, previous research has highlighted the challenges that overseas doctors experience when transitioning to the UK NHS. It may be that UK citizens who have undergone their undergraduate training in other settings are not readily identified as potentially benefiting from additional support. This would be because culturally and linguistically they would not be expected to stand out from home medical graduates and less likely to experience cultural dissonance. As such, policy could highlight this group as one that could be targeted for addition support with transitioning. Additionally, not all doctors entering UK training from overseas have completed their foundation training, these doctors require a supervisor (sometimes from abroad) to verify that they have achieved foundation competencies (the Alternative Certificate of Foundation Competence). At present, it is unclear what proportion of non-UK graduates this relates to and whether they are disadvantaged in any way.

While conducting this study, we noted some inconsistencies in the current UK regulatory policy. For example, European citizens who study outside of the EEA are not expected to sit the PLAB (if having practised in the EEA for at least 3 years) while UK citizens in a similar situation generally would undergo the assessments. In addition, those UK OGs who would usually be expected to demonstrate their competency via the PLAB system did not have significantly better ARCP outcomes than those who did not. Most PLAB candidates eventually pass both parts so the impact of the exam on future ARCP performance in this subgroup may not be substantial.[51] However, previous research has shown that PLAB scores do predict both ARCP and postgraduate exam performance in IMGs.[6 7] Once the UK leaves the EU, there is a potential opportunity for changes to medical regulation. For example, 'Brexit' could potentially allow for the introduction of a national licensing exam that will be taken by all doctors wishing to practice in the UK regardless of nationality or place of qualification, subject to any exemptions that are agreed. The GMC has been consulting on plans to introduce such a test in the form of the UK 'Medical Licensing Assessment'.[52] It is important to point out that in other parts of the world the introduction of such licensing exams do not, in themselves, ensure equivalence in subsequent performance between differing medical graduate groups.[11] Nevertheless, it would hopefully help ensure minimum standards of competence and a greater degree of fairness in the regulatory system.

Those UK nationals who choose to study medicine abroad before returning to the NHS are unlikely to be a

homogenous group. Thus, further research should focus on understanding the qualitative characteristics of this category of doctor. Importantly, we do not know whether, as in the USA, this group of nationals who qualified overseas have inferior patient outcomes compared with other categories of medical practitioner.[14] Certainly, the observations alluded to by McManus and Wakeford,[7] that UK OGs performed more poorly on the knowledge but not the clinical component of the MRCGP, are intriguing. Indeed they may imply, at least for UK OGs who sit the PLAB, that it may be mainly performance on knowledge, rather than skills-based assessments that are at least driving the differential attainment between this latter and other medical graduate groups. Thus, for these reasons research examining actual UK-based clinical practice into differing graduate groups is urgently required.

The present study, in the context of previous work in this area, suggests that the regulations governing the right to practise medicine in a particular country should not be determined by either nationality or place of qualification. Rather they should be based on a reliable and equitable evaluation of clinical ability and other personal qualities essential to the practice of medicine in that specific national context. The introduction of a licensing exam into the UK would provide an opportunity to implement such policy. The impact of such a change should be carefully evaluated as at present it is unclear whether such a universal assessment is likely to translate into improved safety and quality of patient care.[53]

**Acknowledgements** The authors are grateful to GMC colleagues who commented on the paper: Richard Amison, Ben Griffith, Neil Jinks, Andy Knapton, Nicola While and Kirsty White.

**Contributors** PAT led on conception, design, statistical analysis and interpretation of data and is the guarantor of the paper. JO contributed to the cleaning, management and analysis of the data and drafting the article. LWP conducted analyses and produced the data visualisations of the results as well as contributing to drafting and critically appraising the article. DS was involved in cleaning and managing the data as well as drafting, revising the article and critically appraising the content. JJN was involved in providing supervision and advice in relation to the analyses as well as contributing to drafting and critically appraising the article. All authors (PAT, JO, LWP, DS and JJN) have approved the final version of the article submitted.

**Funding** PAT is supported in his research by the National Institute for Healthcare Research (NIHR) via a Career Development Fellowship CDF-2015-08-11 . This paper represents independent research part-funded by the NIHR. The time that DS spent working on this project was funded as part of his role at the General Medical Council (GMC).

**Disclaimer** The views expressed are those of the authors and not necessarily those of the NHS, the NIHR, the Department of Health or the GMC.

**Competing interests** PAT has previously received research funding via a competitive tendering process, as lead and co-applicant from the GMC, Health Education England and the Department of Health for England.

**Patient consent** Not required.

**Ethics approval** The study relied on the analysis of de-identified routinely collected data analysed within a 'safe haven' environment. This was confirmed in writing by the chair of the University of York Department of Health Sciences Ethics Committee.

**Provenance and peer review** Not commissioned; externally peer reviewed.

**Data sharing statement** The data and associated STATA syntax used to manage and analyse the data are archived and may be made available from the GMC on

request within a safe haven environment on an individual basis should a sufficient justification be provided.

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
