## [Reviewer comments · BMJ Open]

ARTICLE DETAILS

TITLE (PROVISIONAL)	UK nationals who received their medical degrees abroad: selection into, and subsequent performance in postgraduate training- a national data linkage study
AUTHORS	Tiffin, Paul; Orr, James; Paton, Lewis; Smith, Daniel; Norcini, John

VERSION 1 – REVIEW

REVIEWER	Duncan Scrmgeour Centre for Healthcare Education and Research Innovation (CHERI), University of Aberdeen, Aberdeen, UK 'None declared'
REVIEW RETURNED	10-Apr-2018

GENERAL COMMENTS	Thank you for the opportunity to review this manuscript. I believe it should be considered for publication with minor revision. This is a useful new analysis of a large dataset comparing the likelihood of success at selection into specialty training for doctors who were UK nationals but obtained their primary medical qualification from outside the UK compared with UK medical graduates. No previous studies have investigated this and the rationale for the study is clearly defined. I am interested to know if the authors included LAT and FTSTA doctors in the study cohort? If so, perhaps these doctors should be removed from the analyses as they do not represent standard training in the UK, and their inclusion creates a more heterogeneous sample. This is avoidable. I have a slight concern with the inclusion of ARCP outcome 5 as an unsatisfactory ARCP outcome. Although the inclusion of this group is unlikely to alter the overall results/conclusions, it may lead to a more meaningful interpretation of the results if they were either excluded altogether or analysed as a separate group i.e. compare unsatisfactory (2,3,4) vs. satisfactory (1 or 6), unsatisfactory vs. insufficient evidence (5) and satisfactory (1 or 6) vs. insufficient evidence (5). There is likely to be a difference between trainees who present insufficient evidence to ARCP panels and those who require extra training/time. It may also help the reader if on p8 line 40 to 46 that the description of ARCP outcomes is changed slightly by mentioning that a less than satisfactory outcome was classified as any outcome other than a "1" and "6". This is clear from Table 3 but I think it should also be in the text, with a description of what outcome 6 is. Perhaps an ARCP table explaining the outcomes would be helpful to those unfamiliar with the process.
--

	It is unfortunate that missing data for selection score and interview score is so vast but the authors have addressed this issue well. However, with the creation of UKMED perhaps more complete datasets will become easier to obtain. Overall I thoroughly enjoyed reading this manuscript. It is a valuable and worthy contribution to the medical education literature.
--	---

REVIEWER	Priya Khanna University of Sydney, Australia None
REVIEW RETURNED	14-Apr-2018

GENERAL COMMENTS	This is a large scale complex yet interesting study with implications for selection and training for a particular cohort of UK nationals who received their medical degree abroad. I have two main concerns: one is that it's relevance for international audience is limited. For instance, it'll be good to describe if these graduates were UK citizen by birth or obtained citizenship after staying in the country for a time specified to obtain citizenship. Also the study is quite dense. It'll be good to describe Introduction part in a more succinct manner, especially the objectives of the study should be defined in a dot points to make it easy for readers to understand the complexity of the study.
---

VERSION 1 – AUTHOR RESPONSE

Reviewer: 1

Reviewer Name: Duncan Scrimgeour

Thank you for the opportunity to review this manuscript. I believe it should be considered for publication with minor revision.

This is a useful new analysis of a large dataset comparing the likelihood of success at selection into specialty training for doctors who were UK nationals but obtained their primary medical qualification from outside the UK compared with UK medical graduates. No previous studies have investigated this and the rationale for the study is clearly defined.

Authors' response: We thanks the reviewer for these positive comments.

Reviewer #1: I am interested to know if the authors included LAT and FTSTA doctors in the study cohort? If so, perhaps these doctors should be removed from the analyses as they do not represent standard training in the UK, and their inclusion creates a more heterogeneous sample. This is avoidable.

Authors' response: This is a valid point. LAT and FTSTA doctors were indeed included. The ARCP outcomes associated with these posts can easily be identified in the dataset as they are coded distinctly. In our analytic dataset doctors who only held LAT/FTSTA doctors made up 2.4% (n=838) of the cohort. This is now mentioned in our methods section. We re-ran an analysis excluding these doctors but the results were not meaningfully altered.

Also, our data supported our experience that it is common practice for some LAT posts to be undertaken as part of a lead up to a substantive training post (that is, many doctors with LAT coded ARCP outcomes also had regular ACRP outcomes recorded subsequently). Thus, they may often represent more standard-type training posts. Thus, we have made a note of this in the limitations section of the discussion but did not feel it was worthwhile making extremely slight changes to the values portrayed in the Tables and the Figures in our results section.

Reviewer #1: I have a slight concern with the inclusion of ARCP outcome 5 as an unsatisfactory ARCP outcome. Although the inclusion of this group is unlikely to alter the overall results/conclusions, it may lead to a more meaningful interpretation of the results if they were either excluded altogether or analysed as a separate group i.e. compare unsatisfactory (2,3,4) vs. satisfactory (1 or 6), unsatisfactory vs. insufficient evidence (5) and satisfactory (1 or 6) vs. insufficient evidence (5). There is likely to be a difference between trainees who present insufficient evidence to ARCP panels and those who require extra training/time.

Authors' response: This is also a valid point. However, to clarify, we did not classify 'outcome 5' as an 'unsatisfactory' outcome as such, but as a 'less satisfactory' or suboptimal one. Using ARCP ratings as an outcome variable has some advantages (that is, almost universally available for doctors in training, regardless of the speciality etc) but some challenges and limitations (the information tends to lie at the lower end of performance in trainees, multi-level structure to the data etc.). Previously, the lead author has therefore conducted extensive exploratory analyses in order to understand out the information from ARCPs can be optimised in such modelling studies. It has been previously been reported that if the ARCP outcomes representing 'extended training time/leave programme' are collapsed then the ARCP outcomes can be treated as ordinal indicators within a multilevel ordinal logistic regression framework. That, is, within analyses, the outcomes conform to the 'parallel odds' assumption that underpin ordinal logistic regression. Please see Supplementary Table 1 within the supplementary and technical appendix to the paper published by Tiffin et al in the BMJ in 2014 which is available at: <https://www.bmj.com/content/348/bmj.g2622> Thus, information on candidates is maximised as additional information is contained within the ranked outcomes. Nevertheless, there is an important point here, The lead author is an ARCP panel member and it is clear that 'an outcome 5' can be, at times, due to insufficient information being provided by third parties, such as a supervisor, rather than the candidate. We have now included a note on this in the potential limitations of our discussion section. However, in the lead author's experience this is in the minority of cases and our previous findings in international medical graduates demonstrated that outcome 5s were associated with, on average, poorer previous performance on the PLAB test. For this reason we believe it should be treated as an intermediate category within the ordinal system implemented,

Reviewer #1: It may also help the reader if on p8 line 40 to 46 that the description of ARCP outcomes is changed slightly by mentioning that a less than satisfactory outcome was classified as any outcome other than a "1" and "6". This is clear from Table 3 but I think it should also be in the text, with a description of what outcome 6 is. Perhaps an ARCP table explaining the outcomes would be helpful to those unfamiliar with the process.

Authors' response: We have amended this description in line with the reviewer comments. In addition we have now modified the ARCP outcomes list, in the text (under the 'Data sources and preparation' section of the 'Methods') and changed it to a bullet points format, accompanied by the original ARCP categories (Note-this does not include the outcomes not associated with training performance (e.g. 'out of programme experience').

Reviewer #1: It is unfortunate that missing data for selection score and interview score is so vast but the authors have addressed this issue well. However, with the creation of UKMED perhaps more complete datasets will become easier to obtain.

Overall I thoroughly enjoyed reading this manuscript. It is a valuable and worthy contribution to the medical education literature.

Authors' response: We agree on the need to work to improve data completeness for recruitment related datasets. With the evolution of UKMED it is hoped data relating to medical selection and regulation will continue to grow. However, it must be noted that it is Oriel who collects the recruitment data rather than UKMED. As the reviewer implies, in this case the use of multiple imputation, as a form of sensitivity analysis, is the optimum way of addressing this issue of missing data in this context. We thank the reviewer for their positive comments.

Reviewer: 2

Reviewer Name: Priya Khanna

This is a large scale complex yet interesting study with implications for selection and training for a particular cohort of UK nationals who received their medical degree abroad.

I have two main concerns: one is that it's relevance for international audience is limited. For instance, it'll be good to describe if these graduates were UK citizen by birth or obtained citizenship after staying in the country for a time specified to obtain citizenship.

Authors' response: We thank the peer reviewer for their positive comments. We would wish to highlight that there is definitely relevance for an international audience. That is we draw parallels with existing evidence from the United States, demonstrating that US citizens who graduate outside of North America are still more likely than non-US citizens who obtain a medical degree abroad to be appointed to medical training posts, despite evidence that they may have poorer patient outcomes than the latter group. Thus there are implications internationally for how selection into such posts is conducted. Unfortunately, we are unable to differentiate between doctors who have UK citizenship from birth, and those who obtain it subsequently. This is because we only had access to data nationality from the list of registered medical practitioners, which reports only nationality at time of registration. This is now mentioned in our limitations section of the discussion.

Also the study is quite dense. It'll be good to describe Introduction part in a more succinct manner, especially the objectives of the study should be defined in a dot points to make it easy for readers to understand the complexity of the study.

Authors' response: In our experience of publishing these types of papers, the introduction needs to be relatively comprehensive in order to communicate the context of medical regulation and selection in the UK for an international readership. However we have now defined the aims of the study more succinctly using bullet points, as suggested by the reviewer.

VERSION 2 – REVIEW

REVIEWER	Duncan Scrimgeour Centre for Healthcare Education and Research Innovation (CHERI), University of Aberdeen. Scotland. None declared
REVIEW RETURNED	18-May-2018
GENERAL COMMENTS	I am very happy with the authors' responses and changes made. I

	would recommend that the manuscript is accepted for publication.
REVIEWER	Priya Khanna University of Sydney, Australia None
REVIEW RETURNED	25-May-2018
GENERAL COMMENTS	The authors have addressed major concerns expressed in the first review.